# Towards an Optimized Ensemble Feature Selection for DDoS Detection Using Both Supervised and Unsupervised Method [note 1]

**DOI:** 10.3390/s22239144

**Published:** 2022-11-25

**Authors:** Sajal Saha, Annita Tahsin Priyoti, Aakriti Sharma, Anwar Haque

**Affiliations:** Department of Computer Science, Western University, London, ON N6A 3K7, Canada

**Keywords:** DDoS, deep learning, ensemble, machine learning, unsupervised model

## Abstract

With recent advancements in artificial intelligence (AI) and next-generation communication technologies, the demand for Internet-based applications and intelligent digital services is increasing, leading to a significant rise in cyber-attacks such as Distributed Denial-of-Service (DDoS). AI-based DoS detection systems promise adequate identification accuracy with lower false alarms, significantly associated with the data quality used to train the model. Several works have been proposed earlier to select optimum feature subsets for better model generalization and faster learning. However, there is a lack of investigation in the existing literature to identify a common optimum feature set for three main AI methods: machine learning, deep learning, and unsupervised learning. The current works are compromised either with the variation of the feature selection (FS) method or limited to one type of AI model for performance evaluation. Therefore, in this study, we extensively investigated and evaluated the performance of 15 individual FS methods from three major categories: filter-based, wrapper-based, and embedded, and one ensemble feature selection (EnFS) technique. Furthermore, the individual feature subset’s quality is evaluated using supervised and unsupervised learning methods for extracting a common best-performing feature subset. According to our experiment, the EnFS method outperforms individual FS and provides a universal best feature set for all kinds of AI models.

## 1. Introduction

Internet Service Providers (ISPs) are experiencing a significant increase in their network traffic in recent years. Such an increase in traffic which is mainly driven by Internet-based applications and digital services often leads to a rise in cyber attacks [1]. Attackers often look for network vulnerabilities and loopholes to gain unauthorized access to the internet infrastructure and services to launch their cyber attacks.

One of the most common cyber attacks in today’s Internet is DDoS (Distributed Denial of Service), which floods a network with unwanted messages, making it impossible for the target server to serve genuine requests from legitimate clients. The recent increase in DDoS attacks has led to an urgent need to take safeguards measures against those attacks. Artificial Intelligence (AI)-based Intrusion Detection Systems (IDSs) have been proposed to guard against such attackers. However, the AI-based IDS requires historical data for learning and detecting attacks in the future. The accuracy of the attack detection models depends on the data quality and complexity. Most of the historical data contains many insignificant features which are irrelevant to the target variable. These redundant features increase the training time of the model and have a detrimental influence on the model’s performance. To address this issue, the Feature Selection (FS) method is used in conjunction with IDS to remove unnecessary features from the dataset, enhancing the model generalization capabilities and minimizing execution time. One of the most critical aspects of AI-based IDS is choosing an optimum feature set that can efficiently classify the target in the dataset and improve accuracy. FS methods not only extract the best features but also increase model interpretability and reduce the overfitting chances of the model [2]. Meanwhile, efficient machine learning applications require FS, as explained in detail with four different types of dataset: (i) conventional data with flat features, (ii) structured features, (iii) heterogeneous data, and (iv) streaming data. Therefore, different FS methods extract a distinct feature set from a dataset, producing diverse detection accuracy for various AI models from supervised and unsupervised learning categories.

In this paper, we investigated and evaluated the performance of an Ensemble Feature Selection (EnFS) approach for both supervised and unsupervised learning models to extract a standard optimal feature set. The EnFS method combines the features selected by the individual FS method to choose the best feature subset from the dataset [3]. In addition, the EnFS technique ensures optimum results in performance and robustness [4]. For our EnFS, we used three primary types of FS techniques such as (i) Filter-based, (ii) Wrapper-based, and (iii) Embedded Filter-based. The Filter-based FS method uses the data properties to select features based on a particular measure. In the Wrapper-based FS method, a model is first trained on a subset of features, and feature selection is dependent on a search criterion in this technique. Then, we determine whether to add or remove features based on the previous model’s conclusions. The Embedded Filter-based FS method combines both the filter and wrapper techniques. Finally, the integrated feature selection approach uses its algorithm to pick the optimal feature subset from the dataset. In this work, we studied fifteen FS methods from three primary FS methods explained above and used their extracted feature sets to train supervised models, including Machine Learning (ML) and Deep Learning (DL) and unsupervised model to identify the best FS method.

The ensemble method works on several techniques: (i) Majority Voting, (ii) Bagging and Posting, (iii) Boosting, (iv) Random Forest, and (v) Stacking [5]. In this paper, we used the Majority Voting (MV) technique for our ensemble method. It uses several feature selection methods to select the dominant features and allocate a vote to each one. Then, it adds up the total votes for each feature and uses the plurality voting technique to choose the optimum features. After that, we compared the detection model performance based on all fifteen individual feature sets, the optimum feature set obtained from EnFS, and the original feature set from our dataset. Our ensemble-based feature set outperforms unique feature sets for both supervised and unsupervised models. We validated the improved performance of the EnFS selection technique using both supervised (ML and DL) and unsupervised models. The current works are primarily focused on implementing ensemble methods only on supervised models such as Bayesian Networks, Regression Trees [6], Support Vector Machine [7], K-Nearest-Neighbors [8], or unsupervised models such as One-Class SVMs (OCSVM) [9,10], Isolation Forest and Local Outlier Factor [11] separately. However, the implementation of all three primary FS methods in both supervised (machine learning and deep learning) and unsupervised models are not reported in the literature. Earlier, we investigated the performance of different FS methods using only supervised models [12]. In this paper, we extended our preliminary work by implementing unsupervised models. In addition, the supervised learning models are enhanced by incorporating more detection mechanisms. The main research contributions of this work are as follow:Implement a wide variety of unique feature selection methods from three major categories: filter-based, wrapper-based, and embedded. We also fine-tuned the hyper-parameters for the feature selection method using the grid search technique. Finally, we compare the performance of individual feature selection methods using stater-of-art machine learning, deep learning, and unsupervised learning models.Ensemble feature sets extracted from individual feature selection methods based on majority voting. Since different feature subset performs differently for a different classification model, we try to combine them to find a better common feature subset for all major types of the attack detection algorithm.Evaluate the performance of the ensemble feature selection method using machine learning, deep learning, and unsupervised learning and compares the performance with the individual feature selection method to extract an optimal feature set.

The rest of the paper is organized as follows: The literature overview of existing intrusion detection systems and related work in machine learning, deep learning, and unsupervised learning is described in Section 2. Next, Section 3 describes the research methodology, including data handling, feature selection, classification model configuration, and experimental environment settings. Next, the experimental result and discussion to compare individual FS method and EnFS method performance is presented in Section 4. Lastly, Section 5 brings our article to a conclusion by drawing some potential research directions.

## 2. Literature Review

Machine Learning is a method by which machines learn how to do tasks without being explicitly programmed to do so each time, resulting in the ability to learn and progress without external intervention. The computer can be trained to be intelligent in various efficient ways, one of which is better interpretations of the inputs extracted by FS methods during the training process. The feature selection is crucial for the machine learning model as a preprocessing step before doing classifications or regression. This technique helps eliminate unnecessary and insignificant attributes from the data collection, intending to improve the model performance.

### 2.1. Supervised Techniques

Tsai et al. [13] provided a review on 55 papers about ML-based intrusion detection and performed a comparative analysis of the existing works based on the classifiers (single, hybrid, and ensemble classifiers) and datasets they used. Mukkamala et al. [14] conducted a comparative study on selective two algorithms such as Artificial Neural Network (ANN) and Support Vector Machine (SVM). They also experimented with an ensemble classification model consisting of ANN and SVM. Their work showed ensemble classification outperformed individual classifiers in intrusion detection. Chebrolu et al. [6] proposed a lightweight intrusion detection technique using ensemble learning and two FS methods. Amiri et al. [7] investigated both linear and non-linear measures to estimate the feature goodness for FS algorithms. They proposed two FS methods called Linear Correlation-based Feature Selection (LCFS) and Modified Mutual Information-based Feature Selection (MMIFS). They evaluated the performance of their proposed FS techniques based on the classification results from the Least Squares Support Vector Machine (LSSVM) model trained by the KDD’99 dataset. Gomes et al. [15] discussed upcoming trends in ensemble learning with the identification of open-sourced tools. In addition, they proposed a taxonomy of ensemble methods for data stream classification. Sagi, Omer, and Lior Rokach [16] reviewed ensemble learning approaches, tools, and techniques. Their research focused on big data compatibility, model transformation, and integration with Deep Neural Networks (DNN) and recommended corresponding popular algorithms. Gao et al. [17] developed an ensemble adaptive voting algorithm for improving detection accuracy using the NSL-KDD dataset. They also summarized a comparative analysis between their proposed Multitree algorithm with existing algorithms. Tu Pham et al. [18] designed an improved IDS using the ensemble feature selection technique, and their model provides high detection accuracy using the NSL-KDD dataset. They used bagging and boosting methods for extracting the optimum feature set from the original dataset. Das et al. [8] claimed an efficient DDoS detection system based on ensemble technique with high accuracy. Their ensemble framework consists of four supervised machine learning classifiers trained on the NSL-KDD dataset. In ref. [19], they present an end-to-end model for cyberattack detection and cyberattack classification using deep learning-based recurrent models. The suggested model retrieves the hidden layer features from recurrent models and then uses a kernel-based principal component analysis (KPCA) features extraction method to choose the best features. An ensemble meta-classifier is then employed to classify the data once the best features from the recurrent models have been combined.

Chandrashekhar and Sahin [20] provided an elaborated study on three primary FS techniques such as filter-based, wrapper-based, and embedded. They analyzed the performance of different feature selection techniques for both supervised and unsupervised learning. Sheikhpour et al. [21] used labeled and unlabeled datasets while exploring the semi-supervised FS method. In their research work, they provided two different hierarchical structured taxonomies for semi-supervised FS methods. Khalid et al. [22] performed a detailed survey on feature selection and extraction techniques to reduce the data dimensionality so that model can learn better from the dataset. They surveyed the existing works on dimensionality reduction techniques used in machine learning and discussed their applicability based on research criteria. Luis et al. [23] proposed an FS algorithm evaluation technique that can estimate the correctness score of a particular FS method with a corresponding explanation. Their evaluation method calculated the algorithm’s score based on relevance, irrelevance, redundancy, and size of the dataset and compared it with related algorithms. The whole simulation was executed precisely in a similar experimental environment. Their study shows that these selective criteria of FSA show firm dependence on data analysis. Adams and Beling [24] demonstrated that FS methods developed for Gaussian Mixture Models (GMM) could be adapted for Hidden Markov Models (HMM) and vice versa. Lin et al. [25] described an efficient intrusion detection mechanism by using the particular advantages of SVM, DT, and Simulated Annealing (SA) and evaluated performance based on the KDD’99 dataset. In the proposed model, SA helps to select the best features from the dataset and optimizes the parameters for SVM and DT. Their model shows the highest accuracy with the minimum number of features in comparison with the existing works. Os-anaiye et al. [26] proposed an ensemble-based multi-iterated FS technique that provides essential features for detecting DDoS attacks in cloud computing. Their feature ensemble framework used four filtered methods: information gain, gain ratio, chi-squared, and ReliefF. The extracted feature by their ensemble method exhibits higher accuracy for the NSL-KDD dataset. Das et al. [27] used the NSL-KDD dataset and provided an ensemble framework for producing optimal feature sets for ML algorithms. They also showed a comparative performance analysis between their work and existing techniques. Dash and Liu [28] extracted features from the dataset by using several feature selection techniques. They trained several ML algorithms such as Naive Bayes (NB), SVM, DT, etc., using extracted features for efficient attack detection.

### 2.2. Unsupervised Techniques

Unsupervised learning is mostly used for exploratory analysis and dimensionality reductions. It can discover the structure in data and can be well used for anomaly detection. The IDS using unsupervised learning has been one of the researched topics lately. Yousef et al. [9]. They stated that the highest overall performance was from the One-class SVM (OCSVM) on the standard Reuters dataset. In comparison to all other algorithms, the OCSVM method is less computationally costly than neural networks. Wang et al. [29] performed research on a hybrid model by using STIDE and Markov Chain kernels, combined with OCSVM. Their proposed method improved classification results and overcame some kernels’ drawbacks such as over-simplicity, overfitting, the requirement of pure normal data, and reliance on the threshold when used standalone. Mhamdi et al. [10] proposed that training with unlabeled or imbalanced data has a high potential for identifying DDoS attacks using SVM. Results show that OCSVM outperforms DL techniques when combined with auto-encoders. Similar research was conducted by Erfani et al. [30] on high-dimensional data set using DBN (Deep Belief Networks) and OCSVM, which showed prominently better results than deep autoencoders. Further, Vasudevan et al. [31] gave a hierarchical approach for constructing a mathematical model that combines unsupervised (Local Outlier Factor (LOF) and OCSVM) and supervised learning techniques. Lazarevic et al. [32] evaluated Local Outlier Factor (LOF), K-Nearest Neighbors (KNN), PCA (Principal Component Analysis), and unsupervised SVM for intrusion detection using the KDD’99 dataset. During the studies of OCSVM, Amer et al. [33] explained SVM-based algorithms had performed reasonably well for unsupervised anomaly detection. Especially the OCSVM is a suitable candidate for investigation when applying unsupervised anomaly detection in practice. LOF is a common density-based method, which is not appropriate for large-scale, high-dimensional datasets because of its high temporal complexity. The increase in computational power of LOF for faster intrusion detection using GPU is explained by Alshawabkeh et al. [34] which showed 100 times speedup in computational time of the system if compared to the CPU results. Karev et al. [35] used Isolation Forest (ISOF) for anomaly detection utilizing HTTP sessions. A two-layer ensemble model using LOF and ISOF is proposed by Cheng et al. [11] for working on skewed datasets and evaluated on factors such as pruning, efficacy, and accuracy. The results outperformed the normal LOF and ISOF results when performed separately. Xiaoling et al. [36] proposed SPIF (Isolation Forest and Spark), which works well with parallelization. Further broadening the aspect into an ensemble learning, Elghazel et al. [37] proposed a novel technique called Random Cluster Ensemble (RCE), which aimed to identify the out-of-bag feature significance from an ensemble of partitions. Both bagging and random subspaces were combined to create an ensemble of component clustering. They further tested RCE using a recursive feature removal strategy on nineteen benchmark datasets and found that it outperformed RCE without RFE (Recursive feature Elimination).

The importance of extracting the best feature set from the original data is evident from the above discussion. However, there are several limitations in the current works. A comparative analysis among existing literature has been summarized in Table 1. Firstly, a limited number of FS algorithms have been considered for the best feature selection. Secondly, a lack of variation in FS method type is apparent in the literature; they used either filter-based, wrapper-based, or embedded FS methods. Although a couple of works consider three major FS categories, the total number of FS methods was significantly lower. In addition, they experimented with only a supervised learning model for performance evaluation. Lastly, maximum works are considered one type of detection model to compare the performance of different FS methods. However, it is inefficient to design an optimum FS method for each model type, such as machine learning, deep learning, and unsupervised learning. In this work, we tried to extend the existing works by incorporating a significantly large set of individual FS methods from three major categories. In addition, the performance of the individual FS methods is evaluated using a wide variety of AI models from both supervised and unsupervised models. Finally, we applied an ensemble technique on individual FS methods to extract a common optimum feature set for all types of the prediction model.

## 3. Methodology

In this section, we describe our methodology, depicted in Figure 1. It consists of three main component: data preprocessing, individual and ensemble feature selection, and attack detection using machine learning, deep learning, and unsupervised learning. We considered UNSW-NB15 dataset for our experiment that is discussed in Section 3.1. However, the dataset requires some preprocessing steps before using them to training our attack detection model, explained in Section 3.2. Then, we describe feature selection component of our methodology in Section 3.3. Next, we summarized models used in the experiment and their evaluation metrics in Section 3.4 and Section 3.5 respectively. Finally, the configuration of our experimental environment is elaborated in Section 3.6.

### 3.1. Dataset

We used the UNSW-NB15 dataset [40] created by the Australian Centre for Cyber Security (ACCS) in the Cyber Range Lab. The IXIA Perfect Storm tool was used to assemble raw network traffic for the UNSW-NB15 dataset. Our experiment focused specifically on DDoS attack classification. All other irrelevant attacks data were removed from the dataset before starting our investigation. Two different dataset configurations were considered respectively for our supervised and unsupervised classification model. The training and testing dataset are almost balanced for our supervised learning models, consisting of 112,001 and 69,996 data instances, respectively. For unsupervised learning, we used 99% benign data and 1% malicious data in the training phase. On the other hand, a random 1000 samples were utilized from the testing dataset to verify our model performance. The overall class distribution of our experimental data is summarized in Table 2.

### 3.2. Data Preprocessing

Data preprocessing is one of the essential steps for designing an efficient machine learning model, and it includes data cleaning, conversion, scaling, and feature extraction. Firstly, the missing values, null values, and infinity values are dropped off from the dataset. Then, we converted categorical features into numerical values so that the machine learning model can process them for future inference. A technique called Label Encoding from the Scikit-Learn [41] library was used to perform the conversion, which replaces each value in the category with a number in a sequence. Finally, a data normalization or feature scaling technique is applied to our dataset, converting all input data to a standard scale. This step is crucial so that the machine learning model might not give more importance to larger range values than smaller ones and make the inference wrong. Therefore, we applied the Min-Max scaling process, which converts each feature to a given range (e.g., 0 to 1).

### 3.3. Ensemble Feature Selection

A dataset consists of *P* columns (features), which describe the behavior of *Q* rows, i.e., data instance in the problem domain, and a target variable *y*. The process of selecting a feature subset p⊆P, which is most essential to classify the target variable, is called Feature Selection (FS). Each FS algorithm selects a different subset of features that give us different accuracy in the machine learning model. So, we used an ensemble-based feature selection method that combines multiple feature subsets extracted by other FS methods to generate an optimal feature subset based on feature ranking [42].

We select a wide variety of feature selection methods from three major categories: Filter-based, wrapper-based, and embedded. A total of 15 different individual feature selection methods have been chosen from these categories. Among them, seven approaches are considered from the filter-based type, and they are Pearson’s correlation (PEARSON), Chi-Square (CHI2), SelectFDR (SFDR), ANOVA, Select Percentile (SPERCENT), SelectFPR (SFPR), and Variance Threshold (VTSLD). Next, we have selected two wrapper-based feature selection methods: Recursive Feature Elimination (RFE) and Mutual Information (MUTINFO) for our experiment. Finally, the rest of the methods are considered from the embedded category. A total of six embedded algorithms are used to extract features in our experiment, and they are Logistic Regression (LRL1), Extra Tree (EXTREES), LASSO Regression (LASSO), Random Forests (RF), univariate linear regression based on *F*-statistics (FREGEX), and Light GBM (LGBM). Each feature selection technique chooses a unique feature set most appropriate for the target categorization. We grid-search different lengths of feature subsets, and in the hyper-parameter configurations of our FS approach, we used a number of the best features, including 10, 15, and 20, and value 20 produced the best results. We then combined all the feature sets chosen by the individual FS approach to create an ensemble feature set. The features from each unique FS method are incorporated into the majority voting (MV) method. The features selected by the MV approach are those that received votes from more than 50% of the potential FS method.

### 3.4. Attack Detection Models

Several machine learning models from both supervised (machine learning and deep learning) and unsupervised categories were implemented to evaluate the performance of EnFS and individual FS methods. There are seven machine learning, four deep learning, and five unsupervised learning models implemented in this experiment to measure and compare the performance based on the various feature sets such as individual feature set, ensemble feature, and original feature set. We used Naive Bayes (NB), Logistic Regress (LR), Neural Network (NN), Decision Tree (DT), Random Forest (RF), Support Vector Machine (SVM), and Stochastic Gradient Descent (SGD) as a supervised machine learning model. In addition, four deep learning models, such as Deep Neural Network (DNN), Convolutional Neural Network (CNN), Long Short Term Memory (LSTM), and Gated Recurrent Unit (GRU), also used as a supervised model to classify the attack. Finally, the DDoS attacks were classified using unsupervised learning models such as One-Class Support Vector Machines (OCSVM), Isolation Forest (ISOF), K Nearest Neighbors (KNN), Average KNN (A_KNN), and Local Outlier Factor (LOF). The hyper-parameters for all classification models are summarized in Table 3. The table contains three separate sections representing Supervised Learning (Machine learning), Supervised Learning (Deep learning), and Unsupervised Learning models parameters used in our experiment. We considered a significant combination of different detection model and feature set to evaluate the performance of feature selection methods. Table 4 summarize the total number of classification models categorized by model category and feature selection type. Total of 119 machine learning models have been considered in our experimentation with five different feature categories. Similarly, we considered total of 68 and 85 deep learning and unsupervised learning model respectively for further analysis.

### 3.5. Evaluation Metrics

We used different evaluation metrics, e.g., Accuracy, Precision, Recall, F-1 Score, and total training time (Time) in second to estimate the performance of our classification models. There are four measurement parameters in the confusion or error matrix shown in Figure 2: True Positive (TP), False Positive (FP), True Negative (TN), and False Negative (FN), which are used to define the evaluation metrics stated above.
(1)$$Accuracy=\frac{TP+TN}{TP+FP+TN+FN}$$
(2)$$Precision=\frac{TP}{TP+FP}$$
(3)$$Recall=\frac{TP}{TP+FN}$$
(4)$$F1-\mathrm{score}=2*\frac{Precision+Recall}{Precision*Recall}$$

Here, the accuracy can be defined as the percentage of true attack detection over total data samples. Precision measures how often the model can correctly identify the DoS attack from the dataset. Reall is the measurement of how many of the DoS samples from dataset the model does distinguish correctly. Finally, F-1 score is the harmonic average of precision and recall.

### 3.6. Software and Hardware Preliminaries

We used Python and the machine learning library scikit-learn [41], Tensorflow-Keras [43] to conduct the experiments in computers with the configuration of Intel (R) i3-8130U CPU@2.20 GHz, 8 GB memory, and 64-bit Windows operating system.

## 4. Results and Discussion

In this section, we analyze and discuss our experimental results. Our experiment is divided mainly into three phases: (i) individual feature selection, (ii) ensemble feature selection and (iii) performance evaluation using supervised and unsupervised models, discussed following.

### 4.1. Individual Feature Selection

In this phase, a wide variety of feature selection methods from three major categories has been selected for the feature extraction from the original dataset. However, the dataset consists of a total of 42 features which are not all contributing significantly to detecting the attack. So It is necessary to select a feature subset that is highly correlated with the target variable. Hence, we experimented with different feature selection methods for a better comparative analysis. However, some feature selection methods, such as Pearson correlation, chi2, accept the length of the expected feature subset as a parameter. Therefore, a grid search technique has been implemented in our experiment to test different feature subset sizes such as 10, 15, and 20 for choosing the optimum feature-length for those feature selection methods. We found 20 is the best-performing parameter for them. The individual result for each feature selection algorithm is summarized in Table 5. According to our experimental results, some feature selection algorithms returns comparatively smaller feature subset. For example, the SPERCENT and EXTREES extracted only 9 and 8 features respectively. However, we only validate the quality of these feature subset when we used them for our attacks classification task. Therefore, we used each feature subset for wide variety of machine learning models training so that we can evaluate their performance better.

### 4.2. Ensemble Feature Selection

After extracting all individual feature subsets, we applied an ensemble technique to find out a subset of features selected by most candidate feature selection methods. A feature is chosen for the ensemble feature set when selected by at least 50% of the individual feature selection methods. Since our ensemble technique is a filter-based approach, we engaged more selectors in the election process for better judgment. For example, our feature selection phase employed 15 different methods. A feature is included in the ensemble feature set when voted by more than or equal to eight individual feature selection methods. Table 6 represents the ensemble feature set with their corresponding total vote as count. The ensemble method reduced the original feature set length by more than 60% for the detection model. From Table 5, we can find some feature selection methods which return smaller feature subset than our ensemble approach. However, the actual performance should be measure with the combination of feature set, corresponding accuracy and execution time. In the next section, we evaluate the performance of individual feature using different models.

### 4.3. Performance Evaluation

We implemented three major types of AI models: machine learning, deep learning, and unsupervised learning to evaluate the performance of the feature selection method to identify any best-performing common feature set for all. Each particular detection model has been trained using 17 different feature sets, including 15 individual feature sets, one ensemble feature, and one original feature set. That indicates a significant combination consisting of various models and feature subsets to evaluate the performance illustrated in Table 4. For example, 119 different (each of seven models trained and evaluated on 17 feature sets) machine learning models were investigated to determine the best performing model and the corresponding feature set. In addition, we analyzed 68 (four models with 17 feature sets) and 85 (five models with 17 feature sets) models, respectively, in deep learning and unsupervised learning. All our experimental results reported in Table 7, Table 8, Table 9 and Table 10 are evaluated based on classification model accuracy, F1-score, precision, recall, and total training time in second (Time).

The best performing machine model for all individual feature sets is summarized in Table 7 where each row represents the performance of a particular FS method and their corresponding machine learning model. According to our experimental results, most of the feature selection method performs well when it combines with the neural network (NN) architecture and it is 47.1% (8 out of 17). The neural network has more generalization capability compared to tree-based algorithms such as decision tree (DT) or random forest (RF) when it trains with larger dataset. However, the NN model is comparatively difficult to find the right hyper-parameters and architecture to avoid the overfitting. In addition, the required time to train the NN model is significantly larger compared to DT or RF as the number of trainable parameters is high for NN. Random forest (RF) algorithms also performs very well with 6 individual feature sets which is more than 35.3% of our feature selection methods. Other 17.6% feature selection methods provide best performance when they trained with decision tree (DT) algorithm. Since RF algorithm consider multiple decision tree to make the final prediction, they generally perform well than DT but it takes more training time. The average training time for NN model is 117.29 s which is the highest among three best classifiers for all individual feature sets while the DT took the lowest average training time of 1.02 s and RF takes 11.09 s.

The ensemble feature set (EN) trained using the neural network (NN) model shows the best performance with an accuracy of 87.2%. The CHI2 feature set provided a similar accuracy of 87.2% but the model training time (180.5 s) is significantly higher (more than double) than the EN feature set training time (78.32 s). The EN feature set provides the overall best performance in terms of accuracy and significantly lower execution time. According to our experimental results, the performance of machine learning model is highly depends on the feature set used to train. For example, the NN model provided best performance when it trains with EN feature set but similar model gave only 71% accuracy with feature set extracted by LASSO method. Table 8 summarizes the best-performing machine learning model with their corresponding feature set. Most of the model (4 out 7) performance (accuracy) is better using the SFPR feature set, although their accuracy is below 80%, which is significantly low in comparison to the best accuracy (87.2%). This result shows that selecting a suitable machine learning model with an efficient feature set is essential for the best performance.

The best performing deep learning model for all particular feature sets is shown in Table 9 where the EN feature set performs better with an accuracy of 86.8 % using the LSTM model. LSTM is one the best performing model considering all individual feature set results but it execution time was comparatively large than other three models. On the other hand, the DNN took lesser execution time but the average prediction accuracy was less than 80%. The number of trainable parameters is considerably huge in LSTM compared to DNN that results in more training time and better accuracy for LSTM. GRU and CNN performance was very close to each other based on number feature sets where they performed well. In addition, their detection accuracy was close and they are respectively 82% and 83% for GRU and CNN. However, the average execution time was almost 67% more in GRU compared to CNN. Table 10 summarizes the best performance (in terms of accuracy) of deep learning models with their respective feature sets. In the case of deep learning models, the EN feature set shows better performance in most cases. It increases the detection accuracy by more than 4% compared with the individual feature set.

The performance of best performing unsupervised learning models for all the feature selection methods is described in Table 11. For example, the A-KNN model with the EN feature set gives us the best accuracy of 76%. The individual best performance for all unsupervised models is shown in Table 12. The accuracy is the same (76%) for both A-KNN and KNN, but the execution time is short in A-KNN. The length of the feature set for KNN is 20 while the ensemble feature set for A-KNN has comparatively smaller set of feature with 13 elements. As a result, we noticed a improved execution time with ensemble feature set with better accuracy. Compared with the individual feature selection method, the EN feature provided a minimum of 8% more accuracy.

A comparison between existing works and our research is summarized in Table 13. All referred results in the comparison table reported the performance on the UNSW-NB15 dataset. Compared to the current work, we performed an extensive analysis of DDoS detection in the same dataset for three main categories of AI models: machine learning (ML), deep learning (DL), and unsupervised learning (UL). The existing works implement one type of detection model at most without any feature selection, while we compared the performance of ensemble feature selection techniques for three diverse models. It is necessary to extract important features for better generalization and running time. That is why we extracted a common best-performing feature set for three different detection models, giving better accuracy and reduced execution time. For example, in [38], they used both ML and DL methods for performance evaluation, but our proposed DL model outperformed by 10% more accurate detection. In the case of ML models, our performance was lower, but it was very close, with a gap of only 3% in the case of model accuracy. In [44], they considered only DL models for DDoS detection with an accuracy of 86% while our proposed model outperformed by 1% with an accuracy of 87%. Compared to the machine learning model in [45], our model accuracy was lower by 4%. However, we could not compare the running time, although it is essential to evaluate the performance.

## 5. Conclusions

Our research explains the importance of selecting an optimal feature set to classify DDoS attacks using AI-based techniques. Existing works analyzed and compared the performance of different FS methods using either machine learning, deep learning, or unsupervised learning. In this work, we performed a comprehensive analysis of a substantial number of FS methods using three main types of AI methods. Our experiment studied 15 individual feature sets, one ensemble feature set, and one original feature set to identify an optimal set of features. The Majority Voting technique ensembles the individual results from different FS methods. It selects features voted by more than 50% of candidate FS methods. The extracted feature sets were used to train seven ML algorithms, four DL models, and five unsupervised models to evaluate the performance of individual FS and EnFS. After analyzing the result, we concluded that our EnFS technique outperforms the FS method in all categories (ML, DL, and UL). However, our results also concluded that only the optimum feature set is insufficient for the best performance. Therefore, the right combination of optimal feature set and classification model is critical for the best classification result. There is some limitation in our proposed methodology. For example, our proposed model should validate with well-known cyber-attack datasets such as NSL-KDD, CICIDS, etc. Furthermore, in the case of feature selection, we grid search the hyper-parameters for the feature selection methods, which we can optimize using dynamic selection for better results. Therefore, We plan to extend our experiment with other well-known datasets in the cyber-security domain as part of our future work. Usage of multiple datasets to validate the approach would be more effective to choose combination of feature selection and detection models. Furthermore, as part of our ongoing research, adversarial machine learning (AML) might be included in the future to maintain the entire system secure while eliminating belligerent adversaries via the safe use of ML techniques in adversarial circumstances.

## Figures and Tables

**Figure 1 sensors-22-09144-f001:**
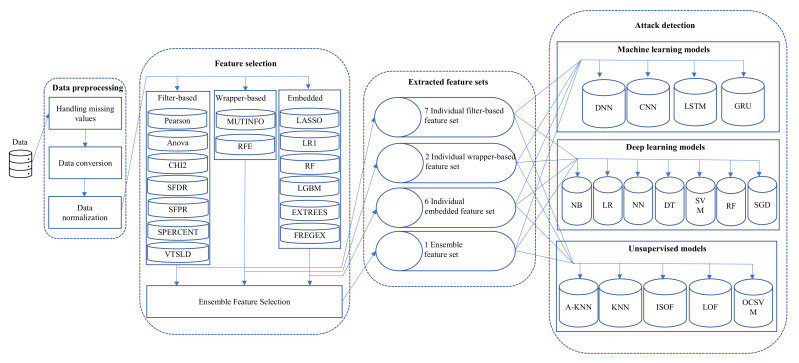
High-level framework of proposed methodology.

**Figure 2 sensors-22-09144-f002:**
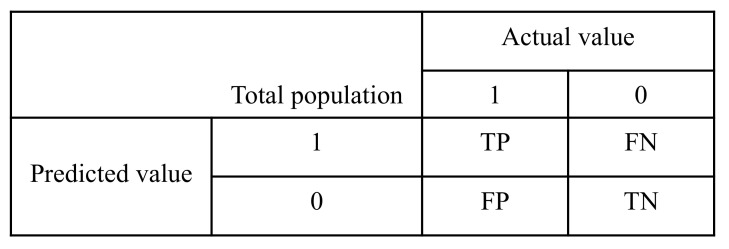
Confusion Matrix.

**Table 1 sensors-22-09144-t001:** Comparative Summary of Existing Works.

Ref.	FS Method	Pros	Cons
[6]	Bayesian networks and the CART	Hybrid architecture involves ensemble and base classifiers for intrusion detection	User to Root(U2R) attack’s were not accurately distinguished.
[7]	Modified Mutual Information-based Feature Selection method (MMIFS)	MMIFS is able to measurea general dependency between features and to rank them.	A huge proportion of DoS and R2L (Root to local)attacks are missed by detection methods.
[8]	Not mentioned	Thorough testing and experiments are carried out to verify the ensemble methods. Their method works well in complex datasets and shows low-time complexities.	Not mentioned
[17]	Not mentioned	Shows betteraccuracy results as compared to other related papers. The generalization effect by gathering advantagesof different algorithms.	Long detection delay in practical application scenarios affects the response time of attack detection.
[18]	Leave-one-out techniques and Naive Bayes classifier, Gain Ratio (GR) technique	The research indicated that used models had high accuracy and low FAR (False Alarm Rate), with the bagging model. They used J48 as the base classifier and worked on a 35-feature subset, producing the best results were 84.25% accuracy and 2.79% FAR.	They performed the comparison only between bagging and boosting ensemble techniques.
[22]	Wrapper and Filter-based methods	Feature selection improves knowledge of the process under consideration, as it points out the features that mostlyaffect the considered phenomenon.The objective of both methods concerns the reduction of feature space in order to improve data analysis.	The computation time of the adopted learning machine and its accuracy need to be considered as they are crucial in machine and data mining applications.
[24]	Gaussian Mixture Models (GMM) and Hidden Markov Models (HMM)	Explored GMMs and HMMs possibilities for supervised and unsupervisedFS methods. Their approach worksbetter with unsupervised learning methods.	GMM related methods were given more emphasis rather than HMM.
[25]	Combination of support vector machine (SVM), decision tree (DT), and simulated annealing (SA)	Generates decision rules to detect new network intrusion attacks.	Detailed comparison with other processes is not visible. Experiments conducted on limited number (DT, SA, SVM) of approaches.
[26]	Info gain, Gain ratio, Chi-squared, ReliefF	Compared to single FS methods, their proposed ensemble-based multi-filter fs selection method shows more efficiency with less complexity	Their process is more prone to false alarm while classification.
[27]	EnFS	Produces an optimal set of features using ensemble technique that improves accuracy significantly. Their technique’s false alarm rate is negligible.	Deep Learning related approaches were not explored.
[37]	RCE and RFE	This research worked to mitigate the gap between ensemble supervised and unsupervised FS learning.	Their proposed method is not very suitable for smaller domains.
[38]	Not mentioned	Proposed a scalable and hybrid noble image processing technique with optimal parameters for both ML and DL architectures.	Training was not conducted on complex DNN architectures.
[39]	Feature reduction	Their approach works for evaluating the shallow and deep networks which were not explored in previous work.	This research did not do experiment analysis for real time deep network data.

**Table 2 sensors-22-09144-t002:** Dataset Summary.

**Supervised Learning**			
	Total instance	Malicious	Benign
Train	112,001	56,001	56,000
Test	69,996	34,998	34,998
**Unsupervised Learning**			
	Total instance	Malicious	Benign
Train	20,000	19,800	200
Test	1000	504	496

**Table 3 sensors-22-09144-t003:** Hyper-parameters setting for classification models.

Model	Parameter Configuration
NB	alpha = 1.0, binarize = 0.0, fitprior = True, classprior = None
LR	randomstate = 0, solver = ‘lbfgs’, multi_class = ‘multinomial’
NN	solver = ‘lbfgs’, alpha = 1 × 10^−5^, hiddenlayersizes = (5, 2)
DT	default parameter
SVM	C = 1.0, kernel = ‘rbf’, degree = 3, gamma = ‘scale’, coef0 = 0.0, shrinking = True, probability = True
RF	default parameter
SGD	max_iter = 1000, tol = 1 × 10^−3^
DNN	No. of hidden layer = 4, No. of neurons = (256,128,64,32), activation = (relu, sigmoid), lr = 0.001, dropout = 0.2, optimizer = adam
CNN	No. of Conv. Layer = 3, No. of neurons in Conv. Layer = (128,64,64), poll_size = 2, kernel_size = 3, No. of dense layer = 4, No. of neurons in dense layer= (256,128,64,32), activation = (relu, sigmoid)
LSTM	No. of hidden layer = 1, No. of neurons = 128, activation = sigmoid, lr = 0.001, optimizer = adam
GRU	No. of hidden layer = 1, No. of neurons = 128, activation = sigmoid, lr = 0.001, optimizer = adam
A-KNN	method=’mean’, contamination = 0.01
ISOF	contamination = 0.01
KNN	contamination = 0.01
LOF	contamination = 0.01
OCSVM	contamination = 0.01

**Table 4 sensors-22-09144-t004:** Total number of models considered for performance evaluation.

Model Type	Filter (7)	Wrapper (2)	Embedded (6)	Ensemble (1)	Original (1)	Total
ML (7)	49	14	42	7	7	119
DL (4)	28	8	24	4	4	68
UL (5)	35	10	30	5	5	85

**Table 5 sensors-22-09144-t005:** Selected features by individual FS method.

Method	Feature Set	Total
PEARSON	[‘service’, ‘stcpb’, ‘sinpkt’, ‘is_sm_ips_ports’, ‘synack’, ‘ct_srv_src’, ‘sload’, ‘dwin’, ‘ct_srv_dst’, ‘tcprtt’, ‘swin’, ‘ct_dst_ltm’, ‘ackdat’, ‘dttl’, ‘dmean’, ‘rate’, ‘dload’, ‘proto’, ‘ct_state_ttl’, ‘sttl’]	20
MUTINFO	[‘dur’, ‘sbytes’, ‘dbytes’, ‘rate’, ‘sttl’, ‘dttl’, ‘sload’, ‘smean’, ‘ct_state_ttl’]	9
SPERCENT	[‘proto’, ‘rate’, ‘sttl’, ‘dttl’, ‘dload’, ‘sinpkt’, ‘dmean’, ‘ct_state_ttl’, ‘is_sm_ips_ports’]	9
CHI2	[‘dur’, ‘proto’, ‘service’, ‘rate’, ‘sttl’, ‘dttl’, ‘sload’, ‘dload’, ‘sinpkt’, ‘swin’, ‘stcpb’, ‘dtcpb’, ‘dwin’, ‘ackdat’, ‘dmean’, ‘ct_srv_src’, ‘ct_state_ttl’, ‘ct_dst_ltm’, ‘ct_srv_dst’, ‘is_sm_ips_ports’]	20
ANOVA	[‘proto’, ‘service’, ‘rate’, ‘sttl’, ‘dttl’, ‘sload’, ‘dload’, ‘sinpkt’, ‘swin’, ‘stcpb’, ‘dwin’, ‘tcprtt’, ‘synack’, ‘ackdat’, ‘dmean’, ‘ct_srv_src’, ‘ct_state_ttl’, ‘ct_dst_ltm’, ‘ct_srv_dst’, ‘is_sm_ips_ports’]	20
FREGEX	[‘proto’, ‘service’, ‘rate’, ‘sttl’, ‘dttl’, ‘sload’, ‘dload’, ‘sinpkt’, ‘swin’, ‘stcpb’, ‘dwin’, ‘tcprtt’, ‘synack’, ‘ackdat’, ‘dmean’, ‘ct_srv_src’, ‘ct_state_ttl’, ‘ct_dst_ltm’, ‘ct_srv_dst’, ‘is_sm_ips_ports’]	20
SFPR	[‘dur’, ‘proto’, ‘service’, ‘state’, ‘spkts’, ‘dpkts’, ‘sbytes’, ‘dbytes’, ‘rate’, ‘sttl’, ‘dttl’, ‘sload’, ‘dload’, ‘dloss’, ‘sinpkt’, ‘sjit’, ‘djit’, ‘swin’, ‘stcpb’, ‘dtcpb’]	20
SFDR	[‘dur’, ‘proto’, ‘service’, ‘state’, ‘spkts’, ‘dpkts’, ‘sbytes’, ‘dbytes’, ‘rate’, ‘sttl’, ‘dttl’, ‘sload’, ‘dload’, ‘dloss’, ‘sinpkt’, ‘sjit’, ‘djit’, ‘swin’, ‘stcpb’, ‘dtcpb’]	20
LRL1	[‘proto’, ‘service’, ‘state’, ‘spkts’, ‘sbytes’, ‘dttl’, ‘dload’, ‘sloss’, ‘djit’, ‘swin’, ‘dwin’, ‘synack’, ‘dmean’, ‘ct_state_ttl’, ‘ct_src_dport_ltm’, ‘ct_dst_sport_ltm’, ‘is_ftp_login’, ‘ct_ftp_cmd’, ‘ct_srv_dst’, ‘is_sm_ips_ports’]	20
LASSO	[‘proto’, ‘state’, ‘dttl’, ‘swin’, ‘tcprtt’, ‘synack’, ‘ct_srv_src’, ‘ct_state_ttl’, ‘ct_srv_dst’, ‘is_sm_ips_ports’]	10
RF	[‘proto’, ‘sbytes’, ‘rate’, ‘sttl’, ‘dttl’, ‘sload’, ‘dload’, ‘tcprtt’, ‘synack’, ‘ackdat’, ‘smean’, ‘dmean’, ‘ct_state_ttl’, ‘ct_dst_src_ltm’, ‘ct_srv_dst’]	16
EXTREES	[‘proto’, ‘sttl’, ‘dttl’, ‘swin’, ‘smean’, ‘ct_srv_src’, ‘ct_state_ttl’, ‘ct_srv_dst’]	8
LGBM	[‘dur’, ‘sbytes’, ‘dbytes’, ‘sload’, ‘sinpkt’, ‘sjit’, ‘dtcpb’, ‘tcprtt’, ‘ackdat’, ‘smean’, ‘dmean’, ‘ct_dst_src_ltm’, ‘ct_srv_dst’]	13
RFE	[‘dur’, ‘proto’, ‘state’, ‘spkts’, ‘sbytes’, ‘dttl’, ‘dload’, ‘sloss’, ‘swin’, ‘dwin’, ‘tcprtt’, ‘synack’, ‘dmean’, ‘ct_srv_src’, ‘ct_state_ttl’, ‘ct_dst_ltm’, ‘ct_src_dport_ltm’, ‘is_ftp_login’, ‘ct_srv_dst’, ‘is_sm_ips_ports’]	20
VTSLD	[‘dur’, ‘proto’, ‘service’, ‘state’, ‘spkts’, ‘dpkts’, ‘sbytes’, ‘dbytes’, ‘rate’, ‘sttl’, ‘dttl’, ‘sload’, ‘dload’, ‘sloss’, ‘dloss’, ‘sinpkt’, ‘dinpkt’, ‘sjit’, ‘djit’, ‘swin’]	20

**Table 6 sensors-22-09144-t006:** Ensemble feature set selected by most the FS method.

Feature	Pearson	MUTINFO	SPERCENT	CHI2	ANOVA	FREGEX	SFPR	SFDR	LRL1	LASSO	RF	EXTREES	LGBM	RFE	VTSLD	Count
dttl	1	1	1	1	1	1	1	1	1	1	1	1	0	1	1	14
proto	1	0	1	1	1	1	1	1	1	1	1	1	0	1	1	13
dload	1	0	1	1	1	1	1	1	1	0	1	0	0	1	1	11
swin	1	0	0	1	1	1	1	1	1	1	0	1	0	1	1	11
sttl	1	1	1	1	1	1	1	1	0	0	1	1	0	0	1	11
ct_state_ttl	1	1	1	1	1	1	0	0	1	1	1	1	0	1	0	11
ct_srv_dst	1	0	0	1	1	1	0	0	1	1	1	1	1	1	0	10
rate	1	1	1	1	1	1	1	1	0	0	1	0	0	0	1	10
sload	1	1	0	1	1	1	1	1	0	0	1	0	1	0	1	10
sinpkt	1	0	1	1	1	1	1	1	0	0	0	0	1	0	1	9
dmean	1	0	1	1	1	1	0	0	1	0	1	0	1	1	0	9
service	1	0	0	1	1	1	1	1	1	0	0	0	0	0	1	8
ct_srv_src	1	0	0	1	1	1	0	0	0	1	1	1	0	1	0	8
is_sm_ips_ports	1	0	1	1	1	1	0	0	1	1	0	0	0	1	0	8
sbytes	0	1	0	0	0	0	1	1	1	0	1	0	1	1	1	8

**Table 7 sensors-22-09144-t007:** Individual FS method best performance summary in machine learning.

FS Method	ML Model	Accuracy	F1 Score	Precision	Recall	Time (s)
PEARSON	NN	0.857	0.870	0.857	0.855	114.30
MUTINFO	RF	0.812	0.852	0.812	0.806	10.50
SPERCENT	RF	0.814	0.855	0.814	0.809	9.10
CHI2	NN	0.872	0.880	0.872	0.871	180.5
ANOVA	DT	0.850	0.863	0.850	0.849	0.80
FREGEX	DT	0.853	0.865	0.853	0.851	0.80
SFPR	NN	0.831	0.840	0.831	0.829	122.81
SFDR	NN	0.831	0.840	0.831	0.829	122.40
LRL1	RF	0.815	0.850	0.815	0.810	9.23
LASSO	NN	0.718	0.725	0.718	0.716	92.04
RF	RF	0.828	0.847	0.828	0.825	11.69
EXTREES	NN	0.838	0.848	0.838	0.837	122.21
LGBM	RF	0.686	0.687	0.686	0.686	15.23
RFE	RF	0.766	0.772	0.766	0.764	10.78
VTSLD	NN	0.806	0.856	0.806	0.799	105.76
ALL	DT	0.844	0.857	0.844	0.842	1.47
EN	NN	0.872	0.879	0.872	0.871	78.32

**Table 8 sensors-22-09144-t008:** Individual machine learning model best performance summary.

ML Model	FS Method	Accuracy	F1 Score	Precision	Recall	Time (s)
DT	CHI2	0.855	0.866	0.855	0.854	0.83
LR	SFPR	0.788	0.847	0.788	0.778	1.43
NB	SFPR	0.670	0.709	0.670	0.655	0.07
NN	EN	0.872	0.879	0.872	0.871	78.32
RF	EN	0.843	0.864	0.843	0.840	9.22
SGD	SFPR	0.786	0.846	0.786	0.776	0.16
SVM	SFPR	0.792	0.853	0.792	0.783	389.51

**Table 9 sensors-22-09144-t009:** Individual FS method best performance summary in deep learning.

FS Method	DL Model	Accuracy	F1 Score	Precision	Recall	Time (s)
PEARSON	LSTM	0.853	0.865	0.797	0.946	424.26
MUTINFO	DNN	0.834	0.854	0.762	0.971	136.94
SPERCENT	DNN	0.803	0.833	0.722	0.986	136.08
CHI2	CNN	0.848	0.863	0.788	0.953	210.07
ANOVA	GRU	0.855	0.867	0.800	0.946	379.14
FREGEX	LSTM	0.856	0.868	0.802	0.946	427.11
SFPR	DNN	0.805	0.835	0.722	0.990	142.55
SFDR	DNN	0.804	0.835	0.721	0.992	134.85
LRL1	GRU	0.803	0.833	0.723	0.982	384.78
LASSO	LSTM	0.757	0.792	0.692	0.927	420.37
RF	CNN	0.840	0.853	0.787	0.932	262.74
EXTREES	CNN	0.826	0.833	0.801	0.869	205.03
LGBM	DNN	0.713	0.698	0.736	0.664	136.19
RFE	LSTM	0.749	0.782	0.690	0.902	462.75
VTSLD	DNN	0.813	0.840	0.732	0.986	145.57
ALL	GRU	0.820	0.843	0.747	0.966	368.68
EN	LSTM	0.868	0.877	0.824	0.937	474.75

**Table 10 sensors-22-09144-t010:** Individual deep learning model best performance summary.

DL Model	FS Method	Accuracy	F1 Score	Precision	Recall	Time (s)
DNN	MUTINFO	0.834	0.854	0.762	0.971	136.94
CNN	CHI2	0.848	0.863	0.788	0.953	210.07
LSTM	EN	0.868	0.877	0.824	0.937	474.75
GRU	EN	0.865	0.874	0.817	0.941	427.58

**Table 11 sensors-22-09144-t011:** Individual FS method best performance summary for unsupervised learning.

FS Method	UL Model	Accuracy	F1 Score	Precision	Recall	Time (s)
PEARSON	KNN	0.73	0.72	0.78	0.73	19.24
MUTINFO	LOF	0.65	0.63	0.70	0.65	1.36
SPERCENT	LOF	0.61	0.60	0.61	0.61	2.11
CHI2	KNN	0.73	0.71	0.81	0.73	24.05
ANOVA	KNN	0.73	0.72	0.78	0.73	18.81
FREGEX	KNN	0.73	0.72	0.78	0.73	27.82
SFPR	KNN	0.68	0.67	0.73	0.68	17.69
SFDR	KNN	0.68	0.67	0.73	0.68	20.65
LRL1	KNN	0.64	0.64	0.65	0.64	35.64
LASSO	KNN	0.63	0.63	0.63	0.63	1.99
RF	A-KNN	0.66	0.64	0.72	0.67	2.30
EXTREES	KNN	0.70	0.67	0.78	0.70	1.92
LGBM	A-KNN	0.51	0.40	0.56	0.52	2.23
RFE	ISOF	0.60	0.53	0.75	0.60	3.21
VTSLD	KNN	0.64	0.64	0.64	0.64	10.64
ALL	A-KNN	0.65	0.62	0.72	0.65	14.09
EN	A-KNN	0.76	0.75	0.80	0.76	2.74

**Table 12 sensors-22-09144-t012:** Individual unsupervised learning model best performance summary.

UL Model	FS Method	Accuracy	F1 Score	Precision	Recall	Time (s)
A-KNN	EN	0.76	0.75	0.80	0.76	2.74
ISOF	LRL1	0.63	0.58	0.76	0.64	5.30
KNN	EN	0.76	0.75	0.81	0.76	2.77
LOF	PEARSON	0.68	0.66	0.73	0.68	14.92
OCSVM	EXTREES	0.55	0.45	0.68	0.55	51.02

**Table 13 sensors-22-09144-t013:** Comparision with the existing works.

Ref.	ML	DL	UL	EnFS	Accuracy
[38]	No	Yes	No	No	ML (90.3%), DL (76.1%)
[44]	Yes	No	No	No	86.04%
[45]	Yes	No	No	No	90.85%
Our work	Yes	Yes	Yes	Yes	ML (87.2%), DL (86.8%), UL (76%)

## Data Availability

UNSW-NB15 dataset at https://research.unsw.edu.au/projects/unsw-nb15-dataset (accessed on 23 October 2022).

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
