# Peer review of "Towards an Optimized Ensemble Feature Selection for DDoS Detection Using Both Supervised and Unsupervised Method†"

_sensors, 2022, doi:10.3390/s22239144_

Round 1

Reviewer 1 Report

1) In the introduction, authors are suggested to introduce the problem, motivate the problem, and summarize the main contributions in detail

2) Figure 1: Deep learning approaches have the capability to extract optimal features. But, authors are using various feature selection methods, Why?

3) The parameter details for all the machine learning and deep learning models can reported. This helps others to reproduce the reported results

4) Compare the performance of the proposed method with other exsisting three methods

5) Literature survey: Recent works can be added in the literature survey and discuss the main limitations of these studies and how the proposed method overcomes these limitations

https://www.sciencedirect.com/science/article/abs/pii/S0045790622004037

6) Discuss the proposed method's advantages and limitations

Author Response

Response to Reviewer Comments

Point 1: In the introduction, authors are suggested to introduce the problem, motivate the problem, and summarize the main contributions in detail

Response 1: Thanks for the comment. We introduced the problem in the second paragraph of the introduction section. And then, we briefly discussed the motivation and our contribution in details in next two sections.

Point 2: Figure 1: Deep learning approaches can extract optimal features. But, authors are using various feature selection methods; why?

Response 2: Thanks for the comment. We agree that the deep learning model can select optimal features. But unnecessary features which do not have a strong correlation with the target can increase model training time. Since deep learning models require high computational power and time compared to traditional machine learning models, we tried to make the training faster by providing a smaller feature subset. 

Point 3: The parameter details for all the machine learning and deep learning models can reported. This helps others to reproduce the reported results

Response 3: Thanks for the comment. The parameter details of all machine learning and deep learning models is reported in Table 4.

Point 4: Compare the performance of the proposed method with other exsisting three methods

Response 4: Thanks for the comment. A comparision among existing work and our proposed work has been summarize in Table 13.

Point 5: Literature survey: Recent works can be added in the literature survey and discuss the main limitations of these studies and how the proposed method overcomes these limitations

https://www.sciencedirect.com/science/article/abs/pii/S0045790622004037

Response 5: Thanks for the comment. We sumarized the pros and cons of existing work in Table 1. Also, the limitation of existing works has been discussed at the end of the Literature Review Section. The paper mentioned here has been added in our literature discussion as per reviwer instruction.

Point 6: Discuss the proposed method's advantages and limitations

Response 6: Thanks for the comment. The advantage and limitation of the proposed work has been discussed in conclusion section.

Reviewer 2 Report

The authors raise an important security topic. They focus in their article primarily on DDOS attacks. The Introduction presented by the authors shows the issues in a satisfactory way by introducing the reader to the subject. The chapter presented to "State of Art" is constructed properly. The authors show sources and refer briefly to them.

The flooded part of this part of the article is the list shown in Table 1. The next chapter is the methodology. Generally, the description of the research itself is appropriate and transparent. The results obtained look promising compared to other solutions.

Technical remarks: Tables from 3 Up have for unknown reasons an annotation at the bottom that it is a table foot (?). Single precision should be introduced in the tables that give a good picture of the results obtained. If a value in a given column is used in style, X.XX, then you cannot write 9.1 and it should be 9.10. This applies to all tables.

The biggest disadvantage of the article that should be corrected is the addition of a discussion in his kind, actual discussion of the results received. What is there is in fact a description of the results presented in the tables. No reflection, references to other tests, etc. It must be supplemented.

Author Response

Response to Reviewer Comments

Point 1: The authors raise an important security topic. They focus in their article primarily on DDOS attacks. The Introduction presented by the authors shows the issues in a satisfactory way by introducing the reader to the subject. The chapter presented to "State of Art" is constructed properly. The authors show sources and refer briefly to them.

Response 1: Thanks for the comment.

Point 2: The flooded part of this part of the article is the list shown in Table 1. The next chapter is the methodology. Generally, the description of the research itself is appropriate and transparent. The results obtained look promising compared to other solutions.

Response 2: Thanks for the comment.

Point 3: Technical remarks: Tables from 3 Up have for unknown reasons an annotation at the bottom that it is a table foot (?). Single precision should be introduced in the tables that give a good picture of the results obtained. If a value in a given column is used in style, X.XX, then you cannot write 9.1 and it should be 9.10. This applies to all tables.

Response 3: Thanks for the comment. We have removed the table note and followed the similar precision style for all tables. 

Point 4: The biggest disadvantage of the article that should be corrected is the addition of a discussion in his kind, actual discussion of the results received. What is there is in fact a description of the results presented in the tables. No reflection, references to other tests, etc. It must be supplemented.

Response 4: Thanks for the comment. We have discussed a comparative analysis between our work and existing works at the end the result and discussion section.

Round 2

Reviewer 2 Report

The authors answered my questions. Thank you.